# Genetic and Phenotypic Parameter Estimates of Body Weight and Egg Production Traits of Tilili Chicken in Ethiopia

**DOI:** 10.3390/ani15182656

**Published:** 2025-09-10

**Authors:** Birhan Kassa, Mengistie Taye, Wondmeneh Esatu, Adebabay Kebede, Mekonnen Girma, Fasil Getachew Kebede, Georgios Banos, Kellie Watson, Olivier Hanotte, Tadelle Dessie

**Affiliations:** 1College of Agriculture and Environmental Sciences, Bahir Dar University, Bahir Dar P.O. Box 5501, Ethiopia; mengistietaye@gmail.com; 2Andassa Livestock Research Center, Bahir Dar P.O. Box 527, Ethiopia; 3Livestock Genetics, Nutrition and Feed Resource (LGNFR), International Livestock Research Institute (ILRI), Addis Ababa P.O. Box 5689, Ethiopia; w.esatu@cgiar.org (W.E.); m.girma@cgiar.org (M.G.); t.dessie@cgiar.org (T.D.); 4Centre for Tropical Livestock Genetics and Health (CTLGH), International Livestock Research Institute (ILRI), Addis Ababa P.O. Box 5689, Ethiopia; o.hanotte@cgiar.org; 5Institute of Biotechnology, Bahir Dar University, Bahir Dar P.O. Box 79, Ethiopia; 6Amahra Regional Agricultural Research Institute, Bahir Dar P.O. Box 507, Ethiopia; adebabayk@yahoo.com; 7Center for Tropical Livestock Genetic and Health (CTLGH), Roslin Institute, University of Edinburgh, Easter Bush Campus, Midlothian EH25 9RG, UK; fasil.kebede@ed.ac.uk (F.G.K.); kellie.watson@ed.ac.uk (K.W.); 8Centre for Tropical Livestock Genetics and Health (CTLGH), Scotland’s Rural College, Animal and Veterinary Sciences, Easter Bush, Midlothian EH25 9RG, UK; georgios.banos@sruc.ac.uk; 9Cells, Organisms, and Molecular Genetics, School of Life Sciences, University of Nottingham, Nottingham NG7 2RD, UK

**Keywords:** egg production, genetic correlation, genetic parameter, growth, Tilili chicken

## Abstract

Indigenous chicken production contributes significantly to the livelihoods of smallholder farmers in Ethiopia. However, the current levels of productivity in terms of body weight and egg number remain low. A breed improvement program for Tilili chicken through selective breeding is underway at Andassa Livestock Center, Ethiopia. In this study, data collected from the breeding program were analyzed to estimate phenotypic and genetic parameters for the body weight and egg production traits. The heritability and genetic correlations of body weight and egg production traits showed that both traits can be improved through selection. Heavier Tilili indigenous chickens tend to lay more eggs over the laying period, making them ideal selection candidates to develop a dual-purpose breed, which is highly desired by the farmers. Our findings reveal the potential of genetically improving the indigenous chicken populations to promote their sustainable utilization and conservation.

## 1. Introduction

African indigenous chickens, kept in low- and medium-input production systems, contribute significantly to the livelihoods of smallholder farmers [1,2]. However, their current productivity levels are too low to meet the growing demands for meat and eggs [3]. Adapted to their local environments, indigenous chickens show high genetic variability within and among populations [4,5], providing opportunities for selective breeding and genetic improvement [5,6,7,8]. In this regard, indigenous chicken genetic resources remained largely underutilized [9,10].

Indigenous chicken faced an increasing threat of genetic erosion due to replacement by commercial breeds, lack of a structured breeding program, and weak conservation strategies [11,12]. In developing countries, 30% of poultry breeds are classified as at risk [11]. These indigenous chickens kept by smallholder farmers have valuable genetic diversity with low institutional support. Moreover, ref. [11] indicated that establishing an integrated national breeding strategy that includes genetic improvement and in situ and ex situ conservation is essential to safeguarding improvement and genetic diversity and ensuring future agricultural resilience.

Selective breeding aims to identify the best-performing individuals and gradually improve the performance of all animals in that breed [13], and it enhances the resilience and productivity of indigenous chickens by improving traits like feed efficiency, growth rate, egg production, and tolerance to environmental stress [14].

The Tilili chicken is among the productive chicken breeds of Ethiopia, predominantly distributed in most parts of West Gojam and Awi Zone, with significant populations in Tilili, Sekela, and Fagita Lekoma districts [3] and raised by smallholder farmers in rural villages with minimal inputs. The population is known for its high phenotypic and genetic diversity, in terms of plumage color, body size, egg laying, and morphological traits [4]. They exhibit diverse plumage, light red in males and partridge or black in females, with black tails in both sexes [15], and under intensive management, Tilili chickens attain an average body weight of 1191 g at 22 weeks of age [4,5]. These characteristics make the Tilili chicken a valuable candidate population (the Tilili chicken population is a local indigenous chicken population found in West Gojam and Awi Zones of western Amhara, managed under traditional backyard systems and known for its adaptation to the local environment and diverse genetic traits) to be considered for a selective breeding program [3].

Genetic and phenotypic parameter estimates guide the setting of breeding objectives and the stage of selection in a breeding program [16]. These estimates are a prerequisite for chicken genetic improvement. Unlike commercial chicken breeds, limited reports are available for some African indigenous chicken populations, and previous genetic parameter estimation studies in African chicken populations target a single selected region [6,16]. Multi-trait selection is influenced by the genetic correlation of traits [17], and moderate to high genetic and phenotypic correlation among the selected traits is required to achieve a significant genetic gain [18]. Unlike Dana et al. [6], Beyihayo et al. [16] reported a negative genetic correlation between body weight and egg number, which is a challenge to run a multi-trait selection program. Heritability of the trait of interest varies with age [19], which makes selection good when heritability is at its peak [20].

Dual-purpose chickens are highly preferred by smallholder farmers in Ethiopia due to their ability to provide both meat and eggs, better adaptability to local conditions (scavenging feed resources, diseases), and lower input requirements. However, their availability remains limited because of insufficient breeding programs, weak multiplication, and dissemination systems [21]. The Tilili chicken breeding program being implemented at Andassa Livestock Research Center has been launched to enhance egg production, body weight, and survival. The present study aims to estimate the heritability and genetic and phenotypic correlations for growth and egg production traits for the Tilili chicken population of Ethiopia. Based on this objective, this study hypothesizes that the Tilili indigenous chicken population possesses sufficient genetic variability for growth and egg production traits, and that selective breeding under intensive conditions can lead to measurable genetic improvement across generations.

## 2. Materials and Methods

### 2.1. Study Site

This study is based on data collected from the Tilili chicken breeding program implemented at the poultry research and multiplication farm at Andassa Livestock Research Center (ALRC), Bahir Dar, in northwestern Ethiopia. ALRC is found 22 km south of Bahir Dar city along the Tis Abay road. It is located between 11°29′ North latitude and 37°29′ East longitude with an elevation of 1730 m above sea level. It receives an average annual rainfall of about 1434 mm, with mean annual temperatures ranging from a maximum of 29.5 °C in March to a minimum of 8.8 °C in January [3].

### 2.2. Breeding Structure

The Tilili chicken breeding program was started in 2021 with an initial flock of mixed sex 3400-day-old chicks (DOC) hatched from 12,500 eggs collected from the local villages of the ecotype. Until 16 weeks of age, both male and female chickens were raised together, after which they were separated. The selection of candidate parents for the next generation was conducted in two stages: the first was based on individual body weight at 16 weeks (large body weight), and the second was on cumulative egg production (larger egg number) up to 24 weeks after the onset of laying. According to the selection schedule, approximately 40 cocks (representing 7–10% of the male population) and 400 hens (50–60% of the females) were selected in each generation based on their body weight performance at 16 weeks of age, specifically favoring individuals with higher body weight. Selected chickens were transferred to the layer selection house and were placed in a pen in a ratio of one cock to ten hens. Pedigree information across generations was recorded. Parentage was assigned through trap nest records, as ten hens mated with a single cock in each pen. Eggs were individually labelled by hen ID for individual data recording and later for hatching.

Individual bird egg numbers were collected up to 44 weeks of age. At the end of week 44, daily collected egg numbers were converted to cumulative 24-week egg numbers, which were used to rank the hens. Accordingly, from the 400 candidates selected based on their 16-week age and weight, 200 hens were selected based on their part-period egg number (the part-period egg number is the cumulative number of eggs laid by a hen during a specific, fixed period after the onset of laying) [22] to become parents for the next generation. Then, after eggs were collected from those 200 hens, they were set for hatching, with four batches to establish the next generation. Hatched chicks were tagged at hatch using wing tags, allowing for precise pedigree tracking and construction of an additive genetic relationship matrix for parameter estimation.

### 2.3. Feeding and Management of Chicken

During the first 3 weeks after hatching, chicks were kept in a deep litter house with 10 chicks per square meter (m^2^) spacing and a 24 h light schedule. Then, the light schedule gradually decreased by an hour per week until it reached 16 h per day. The chicks were provided ad libitum access to starter feed containing 20% crude protein (CP) and 2950–3000 kcal/kg. From 3 to 8 weeks of age, the chicks were fed grower feed containing 18% CP and 2850–2900 kcal/kg. Fresh green legume (Alfalfa; *Medicago sativa*) was fed as supplementation during the starter and grower periods as a source of vitamins and minerals. From 9 to 16 weeks of age, the chicks were given pullet feed containing 16% CP and 2700–2750 kcal/kg. After 17 weeks of age, the chickens were fed layer feed containing 17–18% CP and 2700–2750 kcal/kg at a daily provision of 130 g per chicken.

After 16 weeks of age, the selected candidates of 400 females and 40 males were transferred to layer houses, where they were reared in a floor assigned with a ratio of one male to 10 females in one pen. In each pen, a trap nest was fitted for individual recording of egg number, egg weight, and pedigree. All chickens were vaccinated according to the manufacturer’s guidelines, following a scheduled protocol targeting major poultry diseases including Marek’s disease, Newcastle disease, Infectious Bursal Disease, Fowl typhoid, and Fowl pox (Appendix A).

### 2.4. Traits Measured and Data Used

The dataset used in this study includes two generations, covering the period from 2022 to 2024. In this dataset, body weight was measured at two-week intervals from hatch to 16 weeks of age. Additionally, monthly cumulative egg number and total number of eggs during the first 24 weeks of lay, together with egg weight, were recorded. The weight at hatch was defined as the individual body weight measured at hatch day (BWH, in grams). Subsequently, individual body weight (BW) of chickens was measured at week 2 (BW2), 4 (BW4), 6 (BW6), 8 (BW8), 10 (BW10), 12 (BW12), 14 (BW14), and 16 (BW16) of age. The slight change in sample size over time was due to occasional missing records during weighing and inaccurate chicken ID recording. Some chickens escaped or became mixed with the weighed chickens, making individual identification difficult from the other population.

Individual egg production was recorded daily up to week 44, which was then grouped into monthly egg production periods as 21 to 24 (EPM1), 25 to 28 (EPM2), 29 to 32 (EPM3), 33 to 36 (EPM4), 37 to 40 (EPM5), and 41 to 44 (EPM6). Egg weight was measured weekly. The cumulative of monthly egg production records was used for calculating part-period production. Accordingly, the traits were defined as egg numbers in months 1 to 2 (EPM12), egg numbers in months 1 to 4 (EPM14), egg numbers in months 1 to 6 (EPM16), and average egg weight (AEW). These measures were used to evaluate early and sustained egg production, which is critical for effective selection in indigenous poultry where full-lay records are often impractical [22,23]. Hens without a record for the entire production cycle were excluded from the analysis. The number of animals in the pedigree file that were used in the analysis is presented in Table 1.

### 2.5. Data Analysis

Covariance components and genetic parameters were estimated using a multi-trait animal model with an average information-restricted maximum likelihood (AI-REML) method in WOMBAT software (Version 03-11-2023) [24]. Model convergence was confirmed through the WOMBAT log file, and residuals were inspected with Residuals.dat to search for any indication of non-normality or heteroscedasticity. No major deviation from the model assumption was detected. The statistical models used for the analysis were as follows:*Y* = *Xβ* + *Za* + *Wpe* + *e*
(1)
where

*Y* is the vector of observations for multiple traits;*β* is the vector of observations for multiple traits;*a* is the vector of random direct genetic effects;*pe* is a vector of random permanent environmental effects;*e* is the vector of random residual effects;*X*, *Z*, and *W* are incidence matrices relating observations to fixed and genetic effects and permanent environmental effects (pe) to observations.

Genetic and phenotypic correlations between the traits were estimated using multivariate analysis.

Before conducting the genetic parameter estimation, a General Linear Model (GLM) analysis was performed to identify significant fixed effects. For body weight traits, generation, sex, and hatch/batch were included in the model, while for egg production traits, generation, hatch/ batch, and pen were considered. The result indicates that generation and sex were found to have a significant effect (*p* < 0.05) and were retained as fixed effects (Appendix A). Therefore, generation and sex were fitted as fixed effects for growth traits, while only generation was included as a fixed effect for egg production traits. However, permanent environmental effects (pe) were modeled using repeated records for individual birds across laying months, which helps account for individual-level non-genetic variance. Due to computational limitations associated with the analysis of a large number of traits in multivariate mixed models using WOMBAT, body weight (9 traits: Hatch weight, BW2, BW4, BW6, BW8, BW10, BW12, BW14, and W16) and egg production (10 Traits: EPM1, EPM2, EPM3, EPM4, EPM5, EPM6, EPM12, EPM14, EPM16, and AEW) were analyzed separately. Consequently, nine multivariate trait analyses were conducted for body weight, and ten multivariate trait analyses were conducted for egg production traits. This separation was considered as the number of (co)variance components grows quadratically, leading to a substantial increase in computational time, memory usage, and convergence complexity [24]. However, the genetic and phenotypic correlation was estimated by simultaneously analyzing four body weight traits (BW4, BW8, BW12, and BW16) and four egg production traits (EPM12, EPM14, EPM16, and AEW). BW4, BW8, BW12, and BW16 represent critical growth or developmental stages, and EPM12, EPM14, and EPM16 allow better assessment of genetic relationships across progressively longer production stages, providing insight into selection for both early and extended laying performance in indigenous chickens [6]. Additionally, analyzing cumulative periods reduces the variability seen in individual monthly records, improving the reliability of the genetic correlation estimates.

The heritability was estimated as follows:h^2^ = σ^2^ₐ/σ^2^ₚ(2)σ^2^ₚ = σ^2^ₐ + σ^2^ₚₑ + σ^2^ₑ(3)
where h^2^ is heritability; σ^2^_a_ is the additive genetic variance estimated from *Za* in the model; σ^2^_p_ is the total phenotypic variance, the sum of genetic variance, permanent environmental variance, and residual variance; σ^2^_pe_ is permanent environmental variance estimated from *Wpe* in the model; σ^2^_e_ is residual variance estimated from e in the model.

Genetic correlations and phenotypic correlations were estimated as follows:r_G_ = σ_aij_/√σ^2^_ai_σ^2^_aj_(4)r_P_ = σ_pij_/√σ^2^_pi_σ^2^_pj_(5)
where r_G_ is genetic correlation; r_P_ is phenotypic correlations; σ^2^ₐ is the additive genetic variance; σ^2^_p_ is the total phenotypic variance; σ_aij_ is the additive genetic covariance between traits i and j; σ_pij_ is the phenotypic covariance between traits i and j; σ^2^_ai_ is the additive genetic variance for trait i; σ^2^_aj_ is the additive genetic variance for trait j; σ^2^_pi_ is the phenotypic variance for trait I; σ^2^_pj_ is the phenotypic variance for trait j.

## 3. Results

### 3.1. Growth and Egg Production Performance

The mean, standard deviation, minimum, and maximum values of body weight traits of Tilili chicken obtained in the study are presented in Table 2. Body weight increased with age, from 33.47 ± 3.99 g at hatch to 1189.4 ± 269.32 g at sixteen weeks of age. Likewise, standard deviation also increased with age from hatch weight (3.90) to week 16 body weight (340.7). The coefficient of variation (CV) was lowest for hatch weight (12.89%) and highest for BW12 (34.39%), indicating increasing variability in body weight among individuals as age increases. The growth curve shows a steady increase in body weight from hatch to body weight at 16 weeks of age (Figure 1). The wider error bars on the growth curve indicate that variations among chickens increase with age, reflecting stronger genetic and environmental influences at later ages [25].

The mean monthly and cumulative egg number and average egg weight of Tilili chicken under the selective breeding program are presented in Table 3. The monthly egg number increased as the age of laying hens increased from 6.72 ± 3.33 for EPM1 to 13.08 ± 6.01 for EPM6. The cumulative egg number over 24 weeks of laying of Tilili chicken was 65.89 ± 23.59, with an average egg weight of 45.76 ± 3.03 g.

### 3.2. Variance Components and Heritability of Body Weight and Egg Production Traits

The estimated genetic variance and heritability of growth traits of Tilili chicken under the selective breeding program are depicted in Table 4. Genetic variances for growth traits ranged from 4.71 for hatch weight to 27,765 for BW16. All growth traits exhibit highly significant heritability estimates (*p* ≤ 0.001), indicating that the genetic component contributes substantially to the observed variation in body weight (Table 4; Appendix A). The consistently small *p*-values confirm that the genetic effects are statistically significant across all age-specific body weight traits. Heritability estimates of all growth traits were moderate, ranging from 0.25 ± 0.01 (BW8) to 0.34 ± 0.08 (BW16), reflecting that mass selection is effective for selecting animals that outperform the population for the next generation, and the offspring of selected individuals are likely to have higher body weight at sixteen weeks of age. The heritability estimates of later-age body weights (BW14 and BW16) are slightly higher than the earlier-age body weights (BW2 to BW12), suggesting a stronger genetic influence on body weight as the birds mature (Table 4).

The present study revealed a clear trend in the heritability of body weight traits in relation to the age of the chickens. Heritability estimates increased with age up to 8 weeks and then decreased thereafter, indicating that greater genetic gain may be achieved when selection is performed at later ages, particularly at 16 weeks. Specifically, the heritability estimates for body weight at 2, 4, 6, and 8 weeks were relatively lower compared to those at week 16. This implies that selection based on early-age weight may be less effective due to stronger environmental influences and lower genetic contribution at younger ages.

Genetic and phenotypic variance and heritability estimates of monthly and cumulative egg production and average egg weight of Tilili indigenous chicken in this study are presented in Table 5. The traits EPM5, EPM6, and AEW were statistically significant (*p* ≤ 0.001), reflecting that genetic factors pose a strong influence on the traits. The significance suggests that these traits possess reliable heritability estimates, making them suitable candidates for inclusion in selection programs (Table 5). In contrast, the lower heritability estimates for monthly egg production traits reflect a greater environmental influence, suggesting that improvements in management systems need to be made for the animals to express their genetic potential alongside genetic selection. The heritability estimates of cumulative egg numbers in various ages ranged from 0.08 ± 0.01 for month five (EPM6) to 0.30 ± 0.13 for cumulative egg numbers for the first two months (EPM12). There was no clear trend in heritability estimates of egg number and age of lay. The relatively high (0.36 ± 0.10; *p* < 0.001) heritability estimate for average egg weight (AEW) (Appendix A) suggests good potential for selection response. The lower heritability in EPM6 and no clear trend across egg laying months could be due to the characteristics of the indigenous chicken in irregular laying patterns and laying persistency, and the genetic factor has a limited influence on egg production at this stage. This emphasizes the value of focusing selection by cumulative egg number and when the genetic control is more expressed.

### 3.3. Genetic and Phenotypic Correlation for Growth and Egg Production Traits

For the breeding program, BW16 and cumulative egg number in 24 weeks after the start of egg laying for hens were used as selection criteria. This implies that understanding the association of these traits is important for predicting future growth and integrating selection criteria effectively. The genetic and phenotypic correlations within and among body weight (BW4, BW8, BW12, and BW16) and egg production traits (EPM12, EPM14, EPM16, and average egg weight) are presented in Figure 2.

#### 3.3.1. Genetic Correlations

Body weights at 4, 8, 12, and 16 weeks represent key developmental stages ranging from early growth (BW4) to maturity (market weight) and are commonly used in poultry genetic studies due to their biological relevance and established associations with reproductive performance [26,27]. The genetic correlations among growth traits were positive and significant (*p* < 0.001), ranging from 0.83 to 0.94. This strong correlation confirms that selection based on body weight at 4 weeks of age will positively impact body weight at later ages, implying that selecting growth traits at any stage will achieve genetic progress for other traits. This is expected, as body weight is a continuous trait and tends to be genetically correlated across ages.

The genetic correlations of egg production traits (Figure 2) indicate that selection for increased egg production in one period is likely to result in improvements in other periods. However, most of the genetic correlations between body weight (BW4, BW8, BW12, and BW16) and egg production (EPM12, EPM14, and EPM16) traits were non-significant, suggesting that growth and egg production traits are largely independent at the genetic level. Notably, body weight at sixteen weeks of age (BW16) showed a significant (*p* < 0.001) genetic correlation with egg production during the first four months (EPM1), indicating selecting based on week sixteen body weight might improve egg production.

#### 3.3.2. Phenotypic Correlations

There were positive and significant correlations among growth traits (Figure 2). Specifically, BW16 showed a positive and significant (*p* < 0.001) phenotypic correlation with other growth traits, and with the highest correlation observed with BW12 (0.82). This strong phenotypic correlation supports the idea that body weight is genetically and environmentally consistent across different ages, suggesting that selection for body weight at one age can effectively improve body weight at other ages. The highest significant phenotypic correlation was observed between EM14 and EM16 (r = 0.86; *p* < 0.001), while the smallest was between EM14 and AEW (*p* > 0.05). The large standard errors in phenotypic and genetic correlations, particularly between egg production and body weight traits, suggest limited reliability of the estimates. This may be due to sample size or data structure, emphasizing the need for larger datasets across multiple generations to improve accuracy and strengthen genetic evaluation in Tilili chickens.

Except for the AEW, the phenotypic correlations between body weight and part-record egg number traits were generally low. A moderate phenotype correlation (r = 0.30) was recorded between BW16 and average egg weight (AEW). However, the body weight from week four (BW4) to week twelve (BW12) showed weak negative (*p* > 0.05) phenotypic correlation with cumulative egg number traits (EPM12, EPM14, and EPM16). In contrast, body weight 16 (BW16) showed a slight positive correlation (0.07 to 0.16) with cumulative egg number traits, suggesting that heavier chickens at week 16 may lay more eggs, although the effect is relatively small.

Average egg weight had a significant (*p* < 0.001) correlation with body weight at different ages, ranging from r = 0.13 to r = 30, indicating that heavier birds tend to produce heavier eggs. Overall, the phenotypic correlations between growth and egg production traits suggest that careful consideration is needed when selecting for both traits simultaneously, as selection for growth may not strongly influence egg production and vice versa.

## 4. Discussion

Genetic improvement through the selection of high-performing chickens as replacement stock for the next generation requires population-specific genetic parameter estimates for productive and reproductive traits [16]. Hence, we designed and implemented selective breeding aiming at enhancing live body weight and egg production performance of Tilili chickens.

The mean body weight at different weeks of age in the present study was higher than Horro chickens of central Ethiopia [6]. Similarly, the mean body weight in different weeks of age of the current finding was higher than the values reported for indigenous chicken populations kept under farmers’ management in northwestern Ethiopia [4]. The values obtained in this study also surpassed the values of 25.35, 80.12, 178.32, 281.46, 409.89, and 623.10 for indigenous chicken populations of Uganda at hatch, 2, 4, 6, 8, and 12 weeks of age, respectively [16]. However, the weights at hatch, 4, and 10 weeks of age of the current finding were lower than those of Venda local chickens of South Africa, with values of 34, 286, and 1100 g, respectively [28]. The difference in body weight of different chicken populations could be attributed to differences in genotype, management system, and the nutritional profile of feeds used in local chickens in Africa.

The high standard deviation of the cumulative egg number of indigenous chickens may be attributed to the inconsistent and non-uniform laying performance of local hens, largely attributed to within-population variation. Moreover, the indigenous chicken’s genetic tendency for broody behavior causes large individual variations in laying performance [29].

Cumulative egg number for Tilili chickens at 24 weeks of laying in the present study is higher than Horro chickens [6,30]. However, it is lower than the reported mean cumulative egg number value of 84 eggs for Horro chicken at generation seven [31]. The difference could be attributed to the population differences and the number of generations included in the data analysis.

The present study showed moderate heritability estimates of growth traits ranging from 0.25 ± 0.01 (BW8) to 0.34 ± 0.08 (BW16), which aligns with the other studies [16,18,26,32], as the heritability of body weight traits has moderate to high heritability. This study indicates that 34% of the variation among individuals of week 16 body weight was due to genetic factors, which contributed to selective breeding. Therefore, by selecting animals based on week 16 body weight performance, it is possible to improve the population’s body weight at 16 weeks of age. A similar study has reported that the heritability of body weight in chickens increases with age, with later-age weights (12 to 16 weeks) showing higher heritability estimates than early-age weights, indicating stronger genetic control at maturity [6].

The initial decline in heritability as the birds age could be explained by the diminishing maternal effect over time [26]. As the maternal influence fades, the additive genetic variance becomes more apparent, contributing to the higher heritability observed at later growth stages. In comparison, the heritability estimate for hatch weight in this study was 0.33 ± 0.13, which is slightly lower than the estimate reported for Horro chickens by Dana et al. [6], who found a value of 0.40. This variation may be attributed to differences in breed, environment, or management practices.

The hatch weight and early-age growth rate are highly determined by both the individual’s own genetic potential and the maternal environment [28]. The maternal environment includes egg weight, other quality characteristics, and the incubation process. Heritability values of 0.29 ± 0.01, 0.29 ± 0.08, and 0.28 ± 0.01 in the current study were slightly higher than the values of 0.15, 0.16, and 0.16 for weights at 4, 6, and 8 weeks of age, respectively, estimated for Horro chickens of Ethiopia [6]. However, heritability estimates of body weight at various ages of Tilili chicken in this study were comparable with different African chicken populations. For instance, Wondmeneh et al. [30] reported moderate heritability (0.37) at week 16 body weight for Horro chicken. The heritability estimates in the current study were also in accordance with values reported by Norris and Ngambi [28], in which heritability estimates of weight at hatch, 4, and 10 weeks of age of Venda indigenous chickens were 0.36, 0.25, and 0.4, respectively. Adeyinka et al. [33] reported moderate heritability for body weight at hatch (0.32), week 2 (0.22), week 4 (0.31), week 6 (0.24), and week 8 (0.20). Shad et al. [34] also reported a moderate heritability of 0.25, 0.23, and 0.28 for Iranian Azarbaijan, Esfahan, and Mazandaran ecotypes, respectively. However, Beyihayo et al. [16] reported high heritability estimates of 0.72, 0.67, 0.51, and 0.58 for 2, 4, 8, and 12 weeks of age body weight, respectively, for indigenous chicken populations of Uganda. Similarly, a higher body weight heritability value of 0.55 for Iranian Fars chickens at 12 weeks of age was reported [34]. Osei-Amponsah et al. [19] also reported higher heritability estimates of 0.59, 0.55, and 0.52 for body weight at 8, 10, and 12 weeks for indigenous chickens of Ghana. Cahyadi et al. [35] reported that moderate (0.29) to high (0.63) heritability estimates for body weight traits of Korean native chickens. The discrepancy in heritability estimates of traits reported in different studies is attributed to the method of estimation [36], the number of generations included in the data analysis [37], and breed differences. In general, heritability estimates of growth traits of Tilili chickens were moderate to high and adequate for targeted increased genetic improvement.

The variation in genetic variance across egg production periods (EPM1 to EPM16) reflects the indigenous chicken’s biological characteristics, differences in sexual maturity, low persistency, and broodiness. A higher heritability value in the initial production periods may indicate a higher genetic effect and lower environmental stress, and lower heritability in EPM6 may be indicative of higher biological heterogeneity and reduced genetic control. These findings validate the need for stage-specific selection practices that are in line with the natural laying habit of indigenous chickens. The estimates are in line with the heritability values of 0.32, 0.2, 0.56, 0.24, and 0.35 reported for Horro chicken under an intensive management system [6]. Lower heritability values for the first three months of egg number for Mazandaran (0.16), Azarbaijan (0.01), and Esfahan (0.18) indigenous ecotypes of Iran chicken were reported by Shad et al. [34]. In Nigerian local chickens, the reported heritability for egg number was 0.28 [38]. Beyihayo et al. [16] reported high heritability (0.17) for egg number for indigenous chicken populations of Uganda. The variation in heritability estimates of traits reported in different studies is attributed to the breed differences and method of estimation [36], rearing management [17,39], data size and sampling errors [33], and the number of generations considered and environmental variations [37].

Unlike the present study, a strong genetic correlation of BW16 with EPM12 (0.92), EPM36 (0.69), and EPM16 (0.73) was reported for Horro chicken of Ethiopia [6]. Since chickens in Ethiopia are raised for both meat and egg production, attaining mature body size at earlier ages is a key goal of the production system. Consequently, selection at 16 weeks of age could be the most suitable approach to improve growth [6], which ultimately improves egg production. Notably, a negative genotypic correlation (−0.39) between the egg number and egg weight was reported in local chickens in Nigeria [40], suggesting potential trade-offs between these traits. Most importantly, the correlation between early-age body weight traits and body weight at 16 weeks of age (BW16) increased significantly with age. Dana et al. [6] reported low to moderate genetic correlation among growth traits for Horro chicken of Ethiopia. Adeyinka et al. [33] reported that week 4 body weight has the highest correlation with week 8 body weight (rG = 0.91) for Naked Neck broiler chickens. In the Ugandan indigenous chicken population, the highest genetic correlation (0.80) reported was between BW8 and BW12 [16].

Similar phenotypic correlations for growth traits in African indigenous chicken populations have been reported. For instance, Dana et al. [6] reported low to high phenotypic correlations among growth traits for Horro chicken of Ethiopia. Likewise, Beyihayo et al. [16] reported that body weight at week 4 had a high phenotypic correlation with body weight at week 8 (0.70) for Ugandan indigenous chicken populations. Phenotypic correlation estimates of chicken across studies were varied and inconsistent, attributed to the genetic factors, breed, and management of the chicken kept [6,41]. The inconsistent estimates suggest the importance of quantifying genetic and phenotypic correlations in different laying periods and environments to make effective selection decisions.

Selective breeding relies on phenotypic records and pedigree information to estimate breeding values. On the other hand, genomic selection uses genome-wide markers to predict breeding values at an early age and increase the efficiency of selection, thereby shortening the generation intervals [42,43]. The combined use of phenotypic information, pedigree information, and genotype information can enhance the accuracy of estimated breeding value and accelerate genetic gain for body weight and egg production traits [42,44]. The accuracy of parameter estimates in the present study may have been improved if genomic relationships were used rather than pedigree information [45].

This study focused primarily on egg production and body weight traits and not specifically on the most crucial maternal attributes, such as broodiness, clutch size, and hatchability. The maternal effect can significantly influence reproductive performance and early chicken growth in indigenous chickens. Beyond this, indirect selection based on the simple egg number alone may miss differences in feed conversion and persistence of lay with maternal behavior. While the breeding strategy assumed that high egg-laying hens possess desirable profiles, they were not empirically evaluated. Similarly, this study uses pedigree-based two-generation data with a small number of records for egg production traits. Further analysis, including additional generations and large data sets, is needed to strengthen this finding.

## 5. Conclusions

In this study, we estimated the genetic and phenotypic parameters of indigenous Tilili chickens. Heritability estimates were moderate for body weight traits and low to moderate for egg production traits. BW16 showed a higher level of heritability relative to their respective trait categories, suggesting that selection at these stages of growth will result in genetic progress and respond favorably to selection. Genetic correlations revealed that body weight at 16 weeks of age was highly correlated with other body weight traits. This indicates that the selection of chickens based on week sixteen body weight will also enhance other growth traits. Egg weight showed a moderate genetic correlation with body weight and egg production traits. The significant positive genetic correlation between body weight at various ages and cumulative egg number permits indirect selection and improvement of one trait by the other. However, the small, non-significant phenotypic correlations between growth and egg production traits require attention. Multi-trait selection indices incorporating both body weight and egg production traits may optimize overall performance more effectively than single-trait selection. The parameters estimated in this study could be used along with economic indices for body weight, cumulative egg number, and egg weight traits to develop a selection index aimed at optimizing breeding programs. Overall, this study highlights the potential for simultaneous genetic improvement of growth and egg production traits in indigenous Tilili chickens informed by multi-trait selection strategies. Furthermore, to effectively design indigenous chicken breeding programs, breeders are recommended to place priority on BW16 and key reproductive traits and maintain accurate record-keeping capacity in selection programs.

## Figures and Tables

**Figure 1 animals-15-02656-f001:**
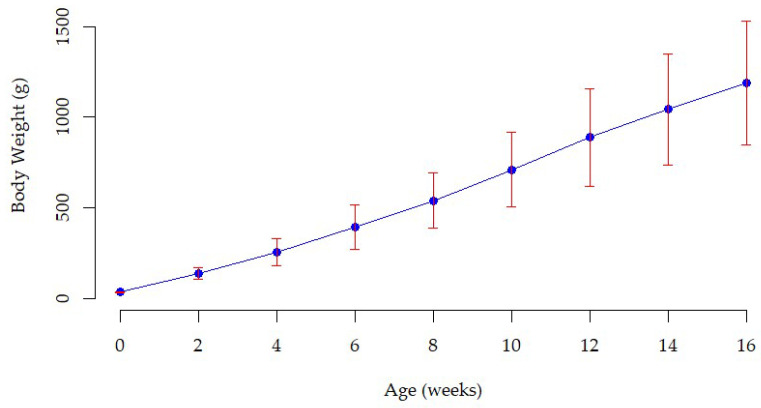
Body weight growth pattern of indigenous Tilili chickens measured from hatch to 16 weeks of age. The vertical bars (red) indicate the standard deviations.

**Figure 2 animals-15-02656-f002:**
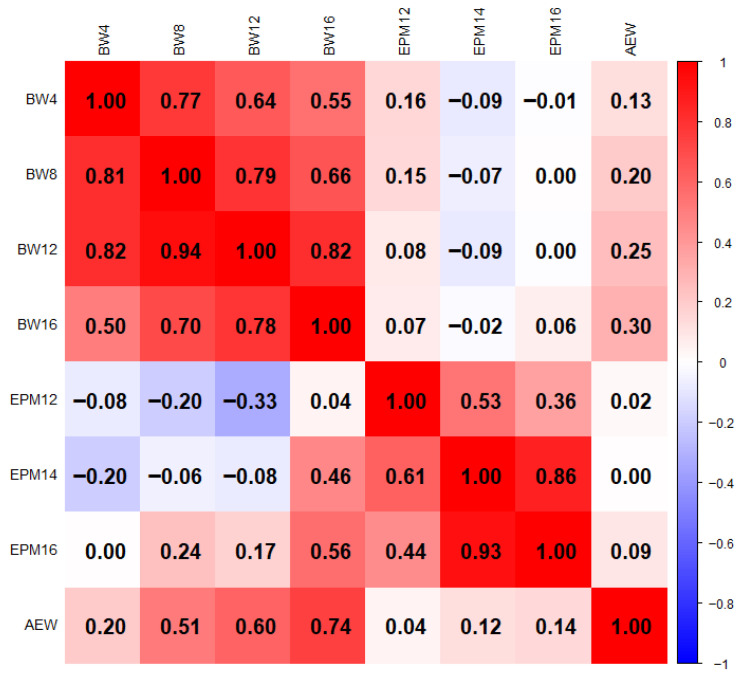
Genetic (below diagonal) and phenotypic correlation (above diagonal) of growth and egg production traits of Tilili chicken under intensive management system. BW4—body weight at 4 week, BW8—body weight at 8 week, BW12—body weight at 12 week, BW16—body weight at 16 week of age, EPM12—egg numbers in month 1 to 2, EPM14—egg numbers in month 1 to 4, EPM16—egg numbers in month 1 to 6, AEW—average egg weight.

**Table 1 animals-15-02656-t001:** Description of the dataset used in the analysis.

Description	Body Weight	Egg Number and Egg Weight
Pedigree Structure for random effect		
Original number of chickens	1393	478
Number of chickens after pruning	1370	473
Proportion (%) remaining	98.49	98.9
Number of sires with progeny	36	31
Number of dams with progeny	140	110
Total Number of chickens with records	1370	473

**Table 2 animals-15-02656-t002:** Mean body weight, standard deviation, and minimum and maximum values of chicken growth traits.

Traits	Records, N	Mean	SD	Minimum	Maximum	CV (%)
HW	1370	33.47	3.90	25	45	12.89
BW2	1282	136.42	32.70	52	250	24.85
BW4	1282	255.74	73.69	75	565	28.87
BW6	1282	392.33	121.93	120	980	30.75
BW8	1279	539.05	153.17	165	1095	29.92
BW10	1274	709.14	205.04	230	1365	31.53
BW12	1274	888.10	269.32	280	1880	34.39
BW14	1260	1042.67	307.16	325	2160	34.25
BW16	1260	1189.94	340.77	340	2190	33.55

HW—hatch weight, BW2—body weight at 2, BW4—body weight at 4, BW6—body weight at 6, BW8—body weight at 8, BW10—body weight at 10, BW12—body weight at 12, BW14—body weight at 14, and BW16—body weight at 16 weeks of age.

**Table 3 animals-15-02656-t003:** Mean monthly, cumulative egg production, and average egg weight of indigenous Tilili chicken under intensive management system.

Trait	Records, N	Mean	SD	Minimum	Maximum	CV (%)
EPM1	473	6.72	3.33	0	22	56.41
EPM2	473	9.96	4.92	0	22	53.75
EPM3	473	11.36	7.02	0	28	66.75
EPM4	473	12.00	5.40	0	25	47.18
EPM5	473	12.35	5.29	0	25	43.18
EPM6	473	13.08	6.01	1	27	42.98
EPM12	473	16.59	6.80	4	36	46.82
EPM14	473	39.86	16.18	6	83	41.61
EPM16	473	65.89	23.59	11	22	35.96
AEW	473	45.76	3.03	39.8	56.8	9.26

EPM1—egg numbers in month 1, EPM2—egg numbers in month 2, EPM3—egg numbers in month 3, EPM4—egg numbers in month 4, EPM5—egg numbers in month 5, EPM6—egg numbers in month 6, EPM12—egg numbers in month 1 to 2, EPM14—egg numbers in month 1 to 4, EPM16—egg numbers in month 1 to 6, AEW—average egg weight.

**Table 4 animals-15-02656-t004:** Variance components and heritability of growth traits of Tilili chicken maintained under selective breeding program.

Trait	N	*σ* ^2^ _g_	*σ* ^2^ _p_	h^2^
HW	1370	4.7171	13.92	0.33 ± 0.13 **
BW2	1282	363.73	1056.5	0.31 ± 0.08 ***
BW4	1282	2007.0	5120.5	0.28 ± 0.01 ***
BW6	1300	5658.0	14,706	0.29 ± 0.08 ***
BW8	1289	10,347	22,578	0.25 ± 0.01 ***
BW10	1274	12,667	38,052	0.31 ± 0.08 ***
BW12	1274	22,103	59,627	0.29 ± 0.07 ***
BW14	1260	23,468	69,470	0.31 ± 0.08 ***
BW16	1370	27,765	79,835	0.34 ± 0.08 ***

HW—hatch weight, BW2—body weight at 2 week, BW4—body weight at 4 week, BW6—body weight at 6 week, BW8—body weight at 8 week, BW10—body weight at 10 week, BW12—body weight at 12 week, BW14—body weight at 14 week, and BW16—body weight at 16 week of age; *σ*^2^_g_—additive variance; *σ*^2^_p_—phenotypic variance; h^2^—heritability. **—statistically significant at *p* < 0.01, ***—statistically significant at *p* < 0.001.

**Table 5 animals-15-02656-t005:** Variance components and heritability estimates of egg production traits of Tilili chicken maintained at a selective breeding program.

Trait	Records, N	*σ* ^2^ _g_	*σ* ^2^ _p_	*h* ^2^
EPM1	473	2.935	9.733	0.30 (0.13) *
EPM2	473	3.041	11.875	0.26 (0.13) *
EPM3	473	2.612	18.266	0.14 (0.11) ^NS^
EPM4	473	2.542	22.334	0.12 (0.11) ^NS^
EPM5	473	7.906	26.536	0.30 (0.12) ***
EPM6	473	2.526	29.901	0.08 (0.01) ***
EPM12	473	9.47	24.91	0.37 (0.12) **
EPM14	473	14.84	88.87	0.18 (0.12) ^NS^
EPM16	473	33.39	259.13	0.13 (0.11) ^NS^
AEW	473	4.34	10.76	0.40 (0.14) ***

EPM1—egg numbers in month 1, EPM2—egg numbers in month 2, EPM3—egg numbers in month 3, EPM4,—egg numbers in month 4, EPM5—egg numbers in month 5, EPM6—egg numbers in month 6, EPM12—egg numbers in month 1 to 2, EPM14—egg numbers in month 1 to 4, EPM16—egg numbers in month 1 to 6, AEW—average egg weight, *σ*^2^_g_—additive variance, *σ*^2^_p_—phenotypic variance, h^2^—heritability. *—statistically significant at *p* < 0.05, **—statistically significant at *p* < 0.01, ***—statistically significant at *p* < 0.001, NS—not statistically significant (*p* ≥ 0.05).

## Data Availability

The raw data supporting the conclusions of this article can be made available by the first author, with due request.

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
