# Peer review of "Genetic and Phenotypic Parameter Estimates of Body Weight and Egg Production Traits of Tilili Chicken in Ethiopia"

_animals, 2025, doi:10.3390/ani15182656_

Round 1
Reviewer 1 Report
Comments and Suggestions for Authors
The manuscript focuses on the estimation of genetic and phenotypic parameters for body weight and egg production traits in selective breeding of Tilili chickens in Ethiopia. The topic aligns with the practical needs of genetic improvement for indigenous African chicken breeds, providing valuable references for the breeding practice of Tilili chickens. The entire text is well-structured with a clear research approach, sufficient and robust data, and core conclusions that are reasonable and persuasive. Overall, it meets the basic requirements for publication. Here, I only suggest that the authors appropriately optimize some details in subsequent revisions to further improve the manuscript.
Major comments:
- Page 3, lines 111-116, Section 2.2, it is mentioned that "approximately 40 cocks and 200 hens were selected as parents based on phenotypic performance". What are the specific quantitative criteria for this phenotypic selection (e.g., body weight thresholds, critical values for egg production, etc.)?
- Page 9, lines 331-336, Could the format of figures (e.g., Figure 6) be further optimized to make the visual presentation more standardized and intuitive?
- Page 13, line 503, the reference section needs to have consistent citation formatting.
Author Response
The comments and reviewer responses are attached in word document.

Reviewer 2 Report
Comments and Suggestions for Authors
Kassa et al. studied genetic and phenotypic parameters for body weight and egg production traits in Tilili indigenous chickens by using a pedigree-based animal model. This work will be helpful in improving dual-purpose poultry breeds in low-input systems like those in Ethiopia. The authors analyzed both within-trait heritability and between-trait genetic/phenotypic correlations. The selective breeding program described in the study is well-structured and contributes to the sustainable conservation and utilization of local poultry germplasm.

Author Response
Dear Reviewer two,
We sincerely thank the reviewer for the constructive and insightful comments. We appreciate your recognition of the scientific merit and relevance of our study. We have carefully revised the manuscript to improve writing clarity, methodological justification, and visual presentation, as suggested. The title has been shortened for clarity, and the abstract now includes details on the breeding structure, sample size, study limitations, and clarified abbreviations. The introduction has been refined to reduce excessive background, and now includes a clear objective, hypothesis, and a concise definition of key terms such as population and breed. Additional descriptions of the Tilili chicken have been incorporated.
In the Materials and Methods section, we clarified the pedigree structure, part-period egg number, and multivariate model use. Details on effects, model assumptions (convergence and residual diagnostics), and the exclusion of fertility and hatchability traits have been added or explained. In the Results section, we included growth curves and heatmaps to support interpretation and clarified reasons for reduced sample sizes. While WOMBAT does not directly provide confidence intervals, we addressed their interpretability based on standard errors and p-values.
In the Discussion, we addressed potential survivorship bias and revised overstatements regarding genetic and phenotypic correlations. We also acknowledged the importance of unrecorded maternal traits (e.g., broodiness, clutch size, feed efficiency) and suggested their inclusion in future studies. Finally, the Conclusion now includes specific recommendations for breeders. References have been updated and formatted consistently, with additional references.
We believe these revisions have significantly improved the manuscript, and we are grateful for the reviewer’s valuable input.
To address your concerns, we have included a detailed point-by-point response outlining the revisions made and highlighted in pink throughout the manuscript. See the attached word word document

Reviewer 3 Report
Comments and Suggestions for Authors
This research provides valuable insights into the genetics of growth and egg production in Tilili chickens, which could also benefit the understanding of other local chicken breeds.
The manuscript is well written overall, with appropriate methodological choices and a clearly structured presentation of results and discussion.
I have a few minor comments:
- The authors should maintain consistency in formatting genetic terms such as variants and heritability using subscripts and superscripts.
- Why were genetic correlations only estimated for EPM12, EPM34, and EPM56? It would be helpful to understand the reasoning behind selecting these time points.
- Table 5 shows a notably low heritability for EPM6, but the manuscript does not address this finding.
- The large differences in genetic variation across traits at different weeks, as shown in Table 5, are quite surprising and merit further discussion.
- The authors reported the average egg weight only as a total value. Was egg weight measured monthly, and if so, were there any differences across months? Clarifying this could enhance the interpretation of the results.
Author Response
Dear Reviewer three,
We sincerely thank you for your constructive comments. We found your suggestions valuable and helpful in further improving the quality of our manuscript. All comments have been addressed and incorporated into the revised version, with changes highlighted in green. The word document is attached
